# Outbreak of Acute Gastroenteritis Caused by Norovirus Genogroup II Attributed to Contaminated Cold Dishes on a Cruise Ship in Chongqing, China, 2017

**DOI:** 10.3390/ijerph15122823

**Published:** 2018-12-11

**Authors:** Li Qi, Xinzhi Xiang, Ying Xiong, Hua Ling, Huangcheng Shen, Wenwen Deng, Wenge Tang, Tao Shen, Qin Li

**Affiliations:** 1Chongqing Municipal Center for Disease Control and Prevention, Chongqing, No. 8, Changjiang 2nd Road, Yuzhong District, Chongqing 400042, China; cdcqili@126.com (L.Q.); cdcxiangxinzhi@126.com (X.X.); xiong-ying-80112@163.com (Y.X.); cdclinghua@126.com (H.L.); qili19812012@126.com (Q.L.); 2Chinese Field Epidemiology Training Program, Chinese Center for Disease Control and Prevention, No. 27, Nanwei Road, Xicheng District, Beijing 100050, China; shentao81@sina.com; 3Zhongxian Center for Disease Control and Prevention, No. 10, Daqiao Road, Zhongxian County, Chongqing 404300, China; shcbaby@163.com; 4Nanan Center for Disease Control and Prevention, No. 410, Yanyu Road, Nanan District, Chongqing 400060, China; dengwenwen1989@126.com

**Keywords:** asymptomatic food handler, cruise ship, case-control study, norovirus, outbreak

## Abstract

In April 2017, several travelers with acute gastroenteritis on a cruise ship were reported. We conducted an investigation to identify the pathogen, mode of transmission, and risk factors. We searched and classified case-patient according to structured case definition, and collect date of the onset, clinical manifestations, and demographic information of cases. A case-control study was implemented to compare foods consumption between cases and asymptomatic travelers. Samples such as feces, rectal swab, vomitus, and environment swab were collected for testing. The attack rate was 18.2% (101/555), four cold dishes served on 11th, April were independently associated with an increased risk of disease: cold potherb (odds ratio (OR): 14.4; 95% confidence interval (CI): 2.2–93.3) and cold garlic sprout (OR: 26.1; 95% CI: 4.9–138.0) served at lunch, cold broad bean (OR: 5.8; 95% CI: 1.3–26.2), and cold cucumber (OR: 13.9; 95% CI: 2.3–84.2) served at dinner. A total of 15 samples were positive for norovirus genogroup II (GII) by using reverse transcription polymerase chain reaction (RT-PCR). This outbreak that occurred on the cruise ship was caused by norovirus GII. The evidence indicated that norovirus was mainly transmitted through consumption of cold dishes on 11th, April, which might be contaminated by asymptomatic cold dish cook.

## 1. Introduction

Norovirus is known to be the major pathogen of acute gastroenteritis worldwide and it causes gastroenteritis outbreaks every year [1,2]. The majority of human norovirus can be classified into two genogroups, I (GI) and II (GII), with the median incubation periods as 1.1 days (95% CI: 1.1–1.2) and 1.2 days (95% CI: 1.1–1.2), respectively [3]. Norovirus infections are generally self-limited with mild to moderate symptoms, although severe morbidity and occasional mortality have been reported in immunocompromised or elderly cases [4]. The symptoms of norovirus infection usually last for 1–3 days [5]. Currently, no vaccines or antiviral therapies are available for preventing and controlling the norovirus infection.

Norovirus is highly infectious with very low infective dose, such as 10 to 1000 viral particles, and it can be transmitted through several routes, such as contact with contaminants, consumption of contaminated food or water, person-to-person transmission via the fecal-oral route, and a particular form of airborne transmission [6,7,8]. Therefore, it has a strong power to trigger epidemic outbreaks in semi-closed and closed settings, such as hospital, nursing home, cafeteria, school, and cruise ship [9,10,11,12,13,14,15,16,17,18,19]. On cruise ships, there are the several suitable conditions for the norovirus outbreaks, such as common food and drinks and high population density. Moreover, it might be a favorable factor for outbreak that most of cruise ship passengers are elderly people, who might be more vulnerable to norovirus infection.

We investigated an outbreak of acute gastroenteritis that was caused by norovirus GII that occurred on a cruise ship sailing on a four-day voyage along the Yangtze River, from Hubei Province to Chongqing Municipality, China, April 2017.

## 2. Material and Methods

### 2.1. Descriptive Epidemiological Investigation

A probable case was defined as any passenger or crew who developed diarrhea (≥3 episodes within 24 h) and/or vomiting (≥2 episodes within 24 h) on the cruise ship during the period of embarkation and disembarkation (10th to 14th, April). While, a confirmed case was defined as any probable case with norovirus nucleic acid positive via the reverse transcription polymerase chain reaction (RT-PCR). An active case finding was conducted among all passengers and crew on the cruise ship by notification and face to face interview. The list of passengers and crew were obtained from the manager of the cruise ship. Each case was face-to-face interviewed by trained public health officers after providing verbal consent. The study was approved by the Ethics Committee of Chongqing Center for Disease Control and Prevention, and as a part of a continuing public health outbreak investigation by Chongqing Municipal Health and Family Planning Commission (Project identification code: k20170042). Information, including demographic information, clinical symptoms, date of onset, and duration of illness was collected through a structured questionnaire. We interviewed the index case, the special cases (asymptomatic food handlers and crew case), and explored the history their activity three days prior to illness onset.

### 2.2. Analytical Epidemiology

Hypothesis was generated based on the time, place, and personal distribution of cases, suspected food items provided by the cruise ship’s Chinese kitchen on 11th April might be the source(s) of infection. In order to test this hypothesis, we performed a case-control study by interviewing cases with inform and consent, and selected asymptomatic passengers randomly to test statistical associations between the exposures and disease (Figure 1). Asymptomatic passengers during the same period were selected as control subjects. The menus were provided by the manager of the restaurant on the cruise ship.

### 2.3. Environmental Hygiene Investigation

We inspected the disinfection facilities and environmental sanitation in the kitchen, restaurant, rooms, and public activity areas of the cruise ship. A further investigation was conducted, including the cold dishes’ preparation process, storage facilities, and disinfection condition, such as used disinfectant, disinfection process of surface, utensils, and dishes, were also inspected and interviewed by the investigation team.

### 2.4. Specimen Collection and Laboratory Tests

Fifteen vomitus, rectal swabs, or feces specimens from cases, twenty-eight rectal swabs from food handlers, six environmental swab samples (kitchen cabinets, rags, fruit knife, and chopping board), two drinking-water samples, and ten remaining food samples from the kitchen were tested for norovirus by reverse transcription-polymerase chain reaction (RT-PCR) in the pathogen laboratory of Chongqing Municipal Center for Disease Control and Prevention (CDC) and Wanzhou CDC, using Daangene^TM^ fluorescence PCR toolkit for norovirus GI and GII,. We used MagNA Pure 96 System (Roche) to do nucleic acid extraction, and did reverse transcription with 1 cycle at temperature of 50 ℃ for 10 min, denaturation with 1 cycle at temperature of 95 ℃ for 10 min, amplification with 45 cycles at temperature of 95 ℃ for 15 s, and did acquisition at the temperature of 58 ℃ for 45 s. The cases provided verbal consent before providing specimen.

Although norovirus was implicated as the etiologic agent based on clinical symptoms specific for oral questions, samples were also tested for intestinal pathogenic bacteria, including *Salmonella, Shigella, enterohaemorrhagic Escherichia coli, Staphylococcus, Bacillus cercus*, and *Vibrio parahaemolyticus,* in accordance with standard isolation protocols at the laboratory of Zhongxian CDC.

### 2.5. Statistical Analysis

All epidemiological and laboratory data were entered into EpiData 3.1 software (Version 3.1, EpiData Association, Odense, Denmark). The distributions of the major symptoms in the outbreak were summarized based on frequencies and percentages. Univariate analysis was conducted to compare the characteristics and exposure of cases and controls using the Chi-square or Fisher’s exact test (when indicated). Variables significant at *p* value < 0.05 in univariate analysis and potential confounders (age, gender) were included in unconditional logistic regression analysis, using a stepwise process with criteria of entry as 0.05 and criteria of excluding as 0.06. All statistical tests were two-tailed and *p* values < 0.05 were considered to be statistically significant.

## 3. Results

### 3.1. Descriptive Epidemiology

The affected cruise ship was sailing along the Yangtze River from Hubei Province to Chongqing Municipality, China, which contains six decks and 225 rooms with independent bathrooms. At the time of this outbreak, there were 132 crew and 423 passengers on board.

A total of 101 acute diarrhea cases met the definition of the probable cases, including 15 cases with laboratory-confirmed norovirus infection. The total attack rate was 18.2% (101/555). Among the cases, 33 were male and 68 were female, with a significant higher attack rate in the females (*χ^2^* = 4.9, *p* < 0.05). The median age of the cases was 64 years old (range: 2–68).

Among the 101 cases, 88 (87.1%) suffered from diarrhea, 75 (74.3%) with vomiting, 42 (41.6%) with bellyache, 29 (28.7%) with nausea, and nine (8.9%) with fever. Twenty-nine cases required hospital admission; however, no severe dehydration or death occurred.

The 101 cases included 100 passengers and one crew, the attack rate was higher in passengers (23.6%) than that in crewmembers (0.8%), (*χ^2^* = 35, *p* < 0.01). According to the distribution of room, the attack rate was the highest on the third deck (33.3%), following by the fifth deck (30.6%), and then the ground floor (28.6%). Significant differences were shown in the attack rates between crew and passengers and among different decks (Table 1).

The first case was reported on 11:00 12th, April. The time distribution showed a peak (71% of the reported cases) between 16:00 on 12th, April to 4:00 on 13th, April (see Figure 2).

The pattern of the epidemic curve indicated a point-source infection. The interviews with the passenger-cases revealed that before embarking on the ship, there was no common food that was eaten by the passengers. The single crew case was with onset date on 13th, who was in charge of cleaning vomitus of passengers.

All passengers had the same traditional Chinese buffet three times a day, while the crew had different meals in different place from the passengers. The buffets were with approximately 18 dishes (hot dishes, cold dishes, and fruit) each meal, while all crew had two or three hot dishes each meal cooked by different cooks.

The passengers and crew drank the same boiled tap water from water tank on the cruise ship.

### 3.2. Analytical Epidemiology

A total of 65 cases and 55 controls were involved in the case-control study. In univariate analysis, cold potherb (odds ratio (OR): 21.8; 95% confidence interval (CI): 7.0–68.1), cold garlic sprout (OR: 9.4; 95% CI: 3.8–23.1) and lettuce salad (OR: 2.6; 95% CI: 1.0–6.9) served at lunch, cold broad bean (OR: 17.9; 95% CI: 5.1–63.1), cold cucumber (OR: 5.8; 95% CI: 2.2–15.5), and chicken with chili (OR: 3.2; 95% CI: 1.3–8.0) served at dinner on 11th, April were associated with illness (Table 2).

All six cold dishes being in significant association with illness in univariate analysis were included in the unconditional logistic regression models. In the final model, four variables were independently associated with an increased risk of disease: cold potherb (OR: 14.4; 95% CI: 2.2–93.3) and cold garlic sprout (OR: 26.1; 95% CI: 4.9–138.0) served at lunch, cold broad bean (OR: 5.8; 95% CI: 1.3–26.2) and cold cucumber (OR: 13.9; 95% CI: 2.3–84.2) served at dinner on 11th, April, adjusted by age and gender.

### 3.3. Environmental Hygiene Investigation

The initial environmental health inspection was conducted on 13th, April, which revealed poor hygiene in the cold dish kitchen, inadequate food storage space in the refrigerator with cooked food mixed with uncooked food, and improper waste disposal.

According to interviews with food handlers, cold dishes were cooked 2–3 h prior to serving without reheating. Ingredients were added by bare hand just before serving. The cook who was in charge of making cold dishes did not wear gloves while working. None of the cooks reported having had acute gastroenteritis symptoms before the outbreak.

### 3.4. Pathogen Detection

#### 3.4.1. Human Microbiological Investigations

Fifteen (14.9%) of 101 cases submitted ether whole stool (*n* = 2), or rectal swabs (*n* = 13), or vomitus (*n* = 1), of which all were positive for norovirus GII. Three (10.7%) of 28 restaurant food handlers submitted rectal swabs tested positive for norovirus GII, including one food handler who prepared the cold dishes and the vegetable salad for passengers, and two food handlers who prepared hot dishes for crew. All of them remained asymptomatic arrived at Chongqing on 14th, April.

#### 3.4.2. Food and Environmental Investigations

One environmental sample (retrieved from chopping board in cold dish kitchen) tested positive for norovirus GII. The food and drinking water samples were negative for norovirus. No bacterial pathogen causing diarrhea was detected in any sample by culture methods.

## 4. Discussion

Our investigation confirmed an acute gastroenteritis outbreak of norovirus GII on a cruise ship in Chongqing, China, in April 2017. Based on the results of epidemiological investigation and laboratory test, we identified that norovirus was transmitted mainly through consumption of cold dishes on 11th, April, which might be contaminated by the asymptomatic cold dish cook on the cruise ship: (1) The epidemic curve (Figure 2) showing that the majority cases had illness onset about 24–28 h after having lunch or dinner on 11th, April. (2) The case-control study indicated four cold dishes served at lunch and dinner on 11th, April were independently associated with an increased risk of disease, which supported a food-borne transmission mode. (3) A total of 15 samples from cases, three samples from asymptomatic food handlers and one swab from chopping board were detected positive for norovirus GII based on RT-PCR. The three asymptomatic food handlers included one cold dish cook and two hot dish cooks. The chopping board was used by the asymptomatic cold dish cook. (4) The outbreak was quickly stopped after the cessation of buffet supply and implementation of thorough disinfection of kitchen.

The asymptomatic food handlers are assumed to have played an important role in the transmission and spread of norovirus in this outbreak, which was also often reported in previous outbreaks [19,20,21,22,23]. It has been shown that norovirus can be transmitted through fingers and fomites [24], and it is relatively resistant and can resist to temperatures as high as 60 degrees. In this outbreak, cold dishes were cooked 2–3 h prior to serving without any reheated process. Ingredients were added by bare hand just before serving without wearing gloves. Hence, it was possible that the asymptomatic cold dish cook had contaminated the chopping board and cold dishes, and passengers became ill after consuming the contaminated cold dishes.

It should be noted that not all the cases had consumed cold dishes served on the cruise ship. The single crew case (attendant) and two asymptomatic hot dish chefs might be infected through person-to-person transmission, given that they lived in the same staff rooms and used a common crew bathroom. Furthermore, some passenger cases might be infected by norovirus after contacting the feces and vomitus contaminated fomites. During the case investigation, we found that most cases did not know that norovirus could be transmitted via daily contact, either that close contact to vomitus or diarrhea was risky.

The outbreaks of norovirus illness on cruise ship could be disruptive and costly; therefore, controlling the outbreak of norovirus infection can be very challenging. On 13th and 14th, April, comprehensive measures were taken to control the outbreak as soon as possible. First, the Chongqing CDC and Zhongxian CDC took measures to “restrict the cases from going outside the room, distribute food to rooms, and close public places such as cinema and gym”, thereby interrupting the close contact among the passengers, so that the number of the cases could be reduced. Second, three asymptomatic food handlers were required to leave the workplace for at least two weeks [25], and all food handlers were asked to reinforce proper hand hygiene through washing hands with soap and water. Third, the Chongqing CDC and Zhongxian CDC disinfected the external environment based on the guideline for environmental cleaning and disinfection of norovirus.

Given the potentially harmful consequences of outbreaks for passengers and crew, the close media spotlight, and the subsequently high cost for cruise ship companies, norovirus outbreaks on the cruise ship represent a serious public health issue. Taking asymptomatic infection among food handlers into consideration is necessary, which makes it difficult to prevent outbreaks by monitoring the health status of food handlers. Therefore, it is crucial for food handlers to strictly comply with personal hygiene and food-safety practices to minimize the risk of transmission of food-borne pathogens. Furthermore, it would be advisable to establish an active reporting system for cruise ship outbreaks, which might help to reduce the burden of imported disease and enable a quick response and minimize the negative effects of increased norovirus activity. Finally, it is necessary to promote awareness campaigns using ship tours, signage, newsletter articles, and in-room televisions to demonstrate the correct behavior that passengers should adopt in order to prevent the spread of norovirus.

Some limitations should be noted in this study. Firstly, we could not identify the sequence homology of norovirus in cases, asymptomatic food handlers, and the chopping board because sequencing was not performed, although all samples were GII positive. Secondly, food handlers might not have disclosed gastroenteritis symptoms to the investigation team because of fear of negative consequences, and how asymptomatic food handlers became infected was not traced. In the future, in-depth interviews on food handlers with reassurance that they will not be fined if they report their illness could facilitate improved symptom reporting. Finally, the number of cases might have been underestimated because the survey was restricted to passengers in a short period before their disembarkation [4,26].

## 5. Conclusions

This gastroenteritis outbreak occurred on a cruise ship was caused by norovirus genogroup II. We speculated that norovirus was mainly transmitted through the consumption of cold dishes on 11th, April, which might have been contaminated by the asymptomatic cold dish cook. This study indicates the importance of preventing food-borne transmission of norovirus on cruise ships. Given the importance of food handlers in the prevention of norovirus infection, it is advisable to reinforce hand hygiene and other standardized practices of food handlers, establish and monitor illness reporting system, and promote health education on crew of cruise ships.

## Figures and Tables

**Figure 1 ijerph-15-02823-f001:**
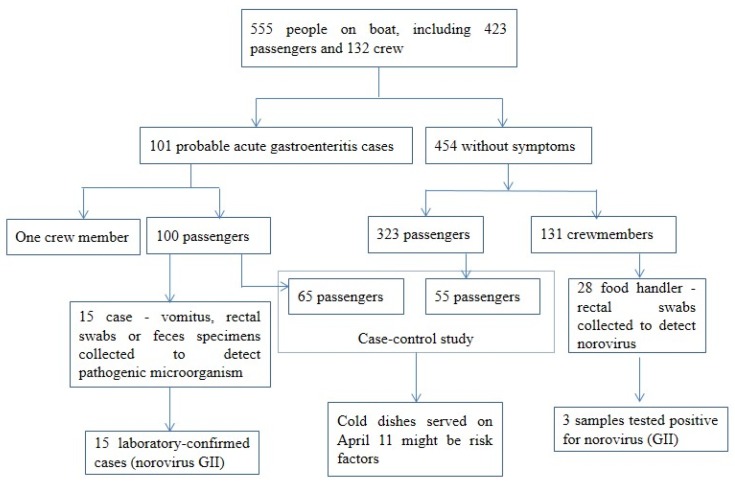
A brief flow-diagram of the norovirus outbreak occurred on cruise ship, Chongqing, China, 2017.

**Figure 2 ijerph-15-02823-f002:**
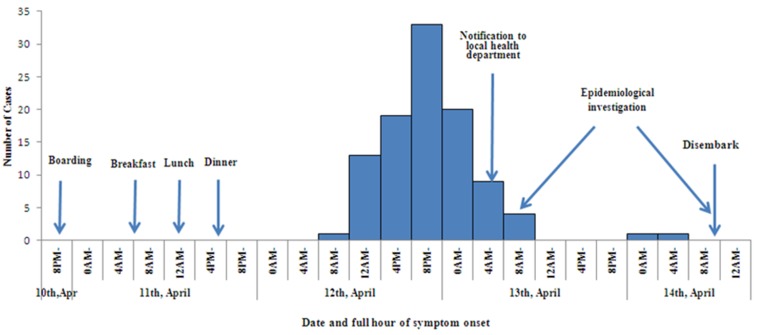
Epidemic curve showing reported cases of the norovirus gastroenteritis on a cruise ship in China, 2017 (by four-hour intervals).

**Table 1 ijerph-15-02823-t001:** Attack rates across different demographics in the norovirus outbreak occurred on the cruise ship, Chongqing, China, 2017.

Variables	Total	Number of Cases	Attack Rate (%)	χ^2^	*p*
Gender					
Male	236	33	14.0	4.9	<0.05
Female	319	68	21.3		
Classification					
Passenger	423	100	23.6	35	<0.01
Crewmember	132	1	0.8		
Deck					
Third deck	153	51	33.3	22	<0.01
Fifth deck	72	22	30.6		
Ground floor	7	2	28.6		
Second deck	52	11	21.2		
Fourth deck	137	15	10.9		

**Table 2 ijerph-15-02823-t002:** Crude odds ratios for statistically significant association between illness and food of the norovirus outbreak on the cruise ship in China, 2017.

Risk Factors		Exposure		%		Odds Ratio (95% CI)
Meals	Food Items	Case (*n* = 65)	Control (*n* = 55)	Case	Control	
Lunch, 11th, April	Cold potherb	41	4	63.1	7.3	21.8 (7.0–68.1)
	Cold garlic sprout	40	8	61.5	14.5	9.4 (3.8–23.1)
	Lettuce salad	18	7	27.7	12.7	2.6 (1.0–2.9)
Dinner, 11th, April	Cold broad bean	33	3	50.8	5.5	17.9 (5.1–63.1)
	Cold cucumber	27	6	41.5	10.9	5.8 (2.2–15.5)
	Chicken with chili	23	8	35.4	14.5	3.2 (1.3–8.0)

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
