# Peer review of "Outbreak of Acute Gastroenteritis Caused by Norovirus Genogroup II Attributed to Contaminated Cold Dishes on a Cruise Ship in Chongqing, China, 2017"

_ijerph, 2018, doi:10.3390/ijerph15122823_

Reviewer 1 Report

Nice descriptive norovirus outbreak report. Although many outbreaks have been documented in the literature, this outbreak investigation was done very well. Suggest to have the lab develop capacity to type the norovirus strains using sequence methods as describes by Cannon JL et al, 2017 [J of Clinical Microbiology]

Author Response

Comment 1. For figure 1, reviewer suggested to change the contents and title.

Response: We revise the figure 1 according to the suggestion as figure 1 showing and the title in line 81.

Comment 2. The reviewer gave suggestions on revising environmental hygiene investigation of method part.

Response: We add some words according to the suggestion, and we clarify the disinfection condition in line 85-88.

Comment 3. The reviewer gave suggestions on revising specimen collection and laboratory tests part.

Response: We provide more information according to the suggestion, in line 90-104.

Comment 4. The reviewer gave suggestions on revising statistical analysis part.

Response: We correct the word according to the suggestion, in line 107.

Comment 5. The reviewer gave suggestions on revising drinking water of passengers and crew members.

Response: We correct the word according to the suggestion, in line 147.

Comment 6. The reviewer gave suggestions on revising title of table 2.

Response: We delete the word “acute gastroenteritis” according to the suggestion, in line 156.

Comment 7. The reviewer gave suggestions on revising environmental hygiene investigation of result part.

Response: We revise our manuscript according to the suggestion in line 165-171.

Comment 8. The reviewer gave suggestions on revising human microbiological investigation of result part.

Response: We revise our manuscript according to the suggestion in line 174-179

Comment 9. The reviewer gave suggestions on revising the discussion part.

Response: We correct the discussion part according to the suggestion in line 228-242.

Comment 10. The reviewer gave suggestions on revising the conclusion part.

Response: We correct some words according to the suggestion in line 248-250.

Reviewer 2 Report

In this manuscript, the authors investigated gastroenteritis outbreak occurred on a cruise ship in China. They found that this outbreak was caused by genogroup II norovirus. Furthermore, norovirus was transmitted through consumption of cold dishes.

The manuscript is clearly written; however, some aspects need clarification.

Specific points;

-It is unclear what genotype of GII norovirus was detected. Please indicate the phylogenetic tree data.

-Please compare sequences of viral genes detected from passengers, food handlers, and an environmental sample. Did they match each other?

-Please clearly describe the method of extracting the viral genes from the sample. In addition, please describe RT-PCR in detail (Temperature conditions, primers, reagents etc.).

Author Response

Comments: The manuscript is clearly written; however, some aspects need clarification. Specific points;

-It is unclear what genotype of GII norovirus was detected. Please indicate the phylogenetic tree data. 

-Please compare sequences of viral genes detected from passengers, food handlers, and an environmental sample. Did they match each other? 

-Please clearly describe the method of extracting the viral genes from the sample. In addition, please describe RT-PCR in detail (Temperature conditions, primers, reagents etc.).

Response: Thanks for your positive comment on our manuscript. As per the suggestion, we clarify this point about sequencing of norovirus in the limitation part in line 236-238. We would like to perform sequencing and phylogenetic tree analysis in future outbreaks investigation.

Regarding the process of RT-PCR, we provide detail information in line 95-99.

Reviewer 3 Report

The manuscript is well written and organized. However, the virus detection method should be better described, indicating the kit companies also and extraction procedure.

Author Response

Comments: The manuscript is well written and organized. The virus detection method should be better described, indicating the kit companies also and extraction procedure.

Comment: Thanks for your positive comment on our manuscript. As per the suggestion, we provide the information in line 95-99.